# Effect of Choline Alphoscerate on the Survival of Glioblastoma Patients: A Retrospective, Single-Center Study

**DOI:** 10.3390/jcm11206052

**Published:** 2022-10-13

**Authors:** Yeong Jin Kim, Tae-Kyu Lee, Myung-Giun Noh, Tae-Young Jung, In-Young Kim, Shin Jung, Kyung-Hwa Lee, Kyung-Sub Moon

**Affiliations:** 1Departments of Neurosurgery, Chonnam National University Research Institute of Medical Science, Chonnam National University Hwasun Hospital and Medical School, Hwasun 58128, Jeollanam-do, Korea; 2Departments of Pathology, Chonnam National University Research Institute of Medical Science, Chonnam National University Hwasun Hospital and Medical School, Hwasun 58128, Jeollanam-do, Korea

**Keywords:** choline alphoscerate, glioblastoma, isocitrate dehydrongenase-wild-type, survival

## Abstract

Cognitive impairment often occurs in glioblastoma (GBM) patients due to the tumor itself and treatment side effects. Choline alphoscerate (L-alpha-glycerylphosphorylcholine, GPC) is frequently used to compensate for cognitive impairment in GBM patients. This study was conducted to determine whether GPC affects the overall survival (OS) and progression-free survival (PFS) of GBM patients. From 2011 to 2020, 187 isocitrate dehydrongenase (IDH)-wild-type GBM patients were analyzed. The patients were classified based on whether GPC was continuously used for at least 3 or 12 months (mos) after GBM diagnosis. Although GPC usage (≥3 mos) did not make significant differences in survival extension, median OS in the long-term GPC group (≥12 mos) was longer with statistical significance, compared to the control group (<12 mos) (38.3 vs. 24.0 mos, *p* = 0.004). In addition to younger age, supratentorial location, complete resection, and MGMT promoter methylation, long-term use of GPC (≥12 mos) was significantly associated with longer OS in multivariate analysis (*p* = 0.019, hazard ratio [HR] 0.532, 95% confidence interval [CI] 0.314–0.900). Despite the limitations of this study, long-term GPC use was possibly associated with prolonged survival in GBM patients. Multi-center prospective randomized studies with a large number of patients are needed to validate these findings.

## 1. Introduction

Glioblastoma (GBM) is a lethal and aggressive malignant brain tumor that is associated with the shortest life expectancy among all human cancers [1]. To improve the survival of patients with GBM, surgery should be performed to remove tumor completely, followed by chemotherapy and radiotherapy [2]. GBM often recurs within one year, and the median overall survival (OS) of GBM patients is typically less than two years after diagnosis. This poor prognosis is inevitable, despite aggressive surgery, full dose of radiation, and treatment with the chemotherapeutic agents, such as temozolomide (TMZ) [3,4]. The invasive infiltration of GBM cells into the surrounding brain tissue is widely responsible for tumor recurrence and the limited effect of current treatments [5]. Several novel therapeutic approaches have been introduced to treat GBM but have not been demonstrated to improve survival in clinical practice [6].

Choline alphoscerate (L-alpha-glycerylphosphorylcholine, GPC) is a choline-containing phospholipid with clinical evidence in the management of cognitive decline in Alzheimer’s disease (AD), cerebrovascular accidents, and aging [7]. GPC, a choline precursor, increases and releases acetylcholine (ACh) to enhance cholinergic transmission [8,9,10]. Since GPC is well tolerated with adequate central nervous system penetration [11], recent investigations have proved that GPC has some positive effects on cognitive improvement in various clinical settings [12,13]. Although the guideline for the pharmacological approach has not been established, many neuro-oncologists have worried about the decline of cognitive function caused by treatments or disease progression and prescribed several drugs [14]. In our country, a recent paper showed that GPC has often been prescribed for various medical situations, including after brain surgery [15]. Many neuro-oncologists in our country have also considered GPC for those with GBM to improve cognitive functioning. However, preclinical studies have shown that GBM cells respond to cholinergic stimulation and acetylcholine receptors are upregulated in GBM-infiltrating lesions [16]. Still, there have been limited studies on the effect of acetylcholine in GBM [17,18]. Furthermore, the increase in choline compounds involved in cell membrane metabolism is a characteristic of malignant glioma [19]. Aberrant choline metabolism in GBM cells is correlated with tumor progression [20]. It is assumed that GPC probably feeds malignant glioma cells as building material to construct cell membranes and progress tumors, which decreases a patient’s life span.

For GBM patients, there is no clinical evidence regarding the survival effect of GPC. Starting with this question, we initiated a retrospective study to evaluate whether there is a survival benefit of GPC in patients with GBM.

## 2. Materials and Methods

### 2.1. Patient Recruitment

From 2011 to 2020, 290 consecutive patients were pathologically diagnosed with GBM in Chonnam National University Hwasun Hospital. The histopathologic diagnosis was established according to the 2016 World Health Organization classifications. The inclusion criteria for this study were as follows: (1) isocitrate dehydrongenase(IDH)-wild-type, primary GBM; (2) adult patients aged >17 years; (3) no other systemic metastatic cancer; and (4) patients had clinical follow-up >three months (mos) and at least one radiologic follow-up. We excluded 36 patients with IDH-mutant type GBM and 41 patients without IDH1/2 exams from this study. Finally, in total, 187 patients (65%) were retrospectively reviewed in this study. This study was approved by the institutional review board of Chonnam National University Hwasun Hospital (IRB No. CNUHH-2022-138). All methods were performed in accordance with the relevant guidelines and regulations.

The dose and frequency of GPC followed the manufactures’ guidelines, generally 400 mg two or three times a day. The patients were classified into a GPC group (continuous usage duration ≥3 mos) or a non-GPC group (<3 mos). As the clinical effect of GPC appears in long-term use (≥12 mos), we performed an additional analysis on patients who were followed for more than 12 months after GBM diagnosis [21]. To verify the long-term effect of GPC, the patients who used GPC for over 12 months were classified into a long-term GPC group. One hundred and twenty-three patients were followed for more than 12 months, of which 38 were in the long-term GPC group and 89 were in the non-long term GPC group.

### 2.2. Surgery and Adjuvant Treatment

Postoperative magnetic resonance imaging (MRI) was performed for all patients within 48 h after surgery. The extent of resection was determined based on the surgical record and was confirmed by an independent radiologist on enhanced MRI. Tumors that did not remain radiographically but were reported as residual tumors during surgery were considered a subtotal resection.

Concomitant chemoradiotherapy (CCRT) with TMZ and following adjuvant TMZ chemotherapy, according to Stupp’s regimen, were used for available patients [22]. The patients received radiation therapy at a total dose of 60 Gy, with daily fractions of 1.8 Gy. Follow-up MRI was performed after CCRT, and then after the third and sixth cycles of TMZ. Patients were followed up weekly during postoperative CCRT, biweekly during adjuvant chemotherapy, and every three months after completion of the standard treatment if the disease was stable. The response assessment in neuro-oncology criteria was used to determine disease progression [23]. If a follow-up MRI revealed disease progression; salvage treatments, including reoperation, re-challenging, or metronomic TMZ; chemotherapy with other agents; and re-irradiation or radiosurgery were delivered according to the treating physician’s discretion.

### 2.3. Statistical Analysis

Categorical variables were compared using the chi-square test or Fisher’s exact test. Continuous variables were compared using Student’s *t*-test, assuming equal variance, and *p*-values were calculated using a two-tailed test. The Mann–Whitney test was used for non-parametric statistics. Survival was analyzed using the Kaplan–Meier method and compared using the long-rank test. OS was the time from initial radiological diagnosis until death. PFS was defined as the time from the first surgery until disease progression (as confirmed by radiologic study) or death. The Cox proportional hazards model with a backward stepwise method was used for multivariate analysis. A *p*-value of <0.05 was considered statistically significant.

## 3. Results

### 3.1. Characteristics of the Patients and Tumors

The patients included 101 men and 86 women. The median age of the patients was 64 (interquartile range [IQR] 54–72) years. The mean preoperative and postoperative Karnofsky performance status (KPS) scores were 80 (standard deviation [SD] ± 13) and 81 (SD ± 10), respectively. The median tumor volume was 31.6 cm^3^. The tumors were located in the supratentorial region (90%) and contacted with ventricles (64%). There were 79 patients (42%) who underwent gross total resection. MGMT promoter methylation status was available for 172 (94%) patients. Among them, 97 patients (56%) showed methylation of the MGMT promoter. A total of 130 patients (70%) had been treated with CCRT. For the entire cohort (187 patients), the median OS and PFS were 15.5 (95% confidence interval [CI] 14.1–17.0) and 7.2 (95% CI 5.8–8.6) months, respectively. A total of 154 patients (82%) died and tumor relapse was observed in 177 (94%) patients by the time of analysis. Patient demographics are summarized in Table 1.

We assessed the difference in clinical characteristics between GPC users (*n* = 80, 43%) and non-users (*n* = 107, 57%) to adjust for confounding factors that may have influenced their prognosis (Table 1). The age, gender, tumor volume, extent of resection, and MGMT promoter methylation status were not significantly different between the two groups. The GPC group showed higher postoperative KPS scores than the non-GPC group. Furthermore, the GPC group was more likely to be treated with CCRT than the non-user group. No patients reported serious side effects or overdose symptoms of GPC.

### 3.2. Survival Outcome

The median OS in the GPC group was longer than that of the non-GPC group [16.1 mos (95% CI 14.6–17.6) vs. 14.2 mos (95% CI 11.8–16.6)]. The corresponding median PFS was 8.4 mos (95% CI 6.6–10.1) and 6.5 mos (95% CI 5.4–8.0) (Figure 1A). The differences in both OS and PFS between the two groups were statistically insignificant (*p* = 0.158 in OS, *p* = 0.092 in PFS), respectively. Analysis of the variables that could be correlated with survival outcome is shown in Table 2. The use of GPC (≥three mos) was not a significant prognostic factor for OS and PFS in the univariate and multivariate analysis.

To verify the long-term effect of GPC, further analysis was performed on 123 patients who were followed for more than 12 months after initial diagnosis (*n* = 34 in long-term user, *n* = 89 in non-long term user). Among several demographic variables, CCRT was frequently adopted in the long-term GPC group (97% vs. 81% in non-long term used group, *p* = 0.042) (Table 3). Median PFS in the long-term users was 14.6 mos (95% CI 9.2–20.0), which was slightly higher than the non-long term users [vs. 9.1 mos (95% CI 7.2–11.0), *p* = 0.082]. Median OS in the long-term GPC group was significantly longer than that of the non-long term GPC group [38.3 mos (95% CI 27.5–49.1) vs. 24.0 mos (95% CI 15.7–32.3), *p* = 0.004] (Figure 1B).

In the multivariate analysis, younger age (*p* < 0.001; hazard ratio [HR] 1.041; 95% CI 1.020–1.064), supratentorial location (*p* = 0.014; HR 0.304; 95% CI 0.118–0.787), complete resection (*p* = 0.001; HR 0.454; 95% CI 0.289–0.715), MGMT promoter methylation (*p* = 0.01; HR 0.525; 95% CI 0.321–0.859), and long-term use of GPC (≥12 mos) (*p* = 0.019; HR 0.532; 95% CI 0.314–0.900) were independent good prognostic factors for OS (Table 4). CCRT was the only independent prognostic factor of PFS (*p* = 0.003; HR 0.445; 95% CI 0.263–0.752), whereas long-term use of GPC was not (*p* = 0.358).

## 4. Discussion

This study was designed to investigate the effect of GPC on survival outcome in patients with GBM. Standard treatment for GBM includes maximal safe surgical resection followed by CCRT and adjuvant chemotherapy using TMZ. Despite aggressive therapies, the prognosis remains dismal and novel approaches are required to improve survival outcomes. Although the investigation for the effect of GPC on brain tumors is lacking, many neuro-oncologists have used GPC to improve patient cognition. Cognitive impairment associated with the brain tumor and the sequelae of its treatments occurs in 50–90% of patients and can reduce the quality of a patient’s life [24,25]. Therefore, many patients are administered GPC after the initial diagnosis or surgery, or with chemotherapy and radiotherapy. Hence, a clinical study is required to identify the effect of GPC in patients with GBM. 

GPC is widely administered for a long time. Adverse effects and drug interactions have been rare in more than 30 years of clinical experience [26,27,28,29]. GPC, a choline-containing phospholipid, provides both free choline and phospholipid to synthesize Ach and to construct nerve cell membranes in the brain [11]. GPC is still used in the treatment of degenerative brain diseases, such as AD and vascular dementia. Unlike a degenerative disease, GBM is an uncontrolled cell proliferative disease. Theoretically, its properties might cause tumor progression. The biosynthesis of tumor cell membranes is rapid due to their fast proliferation. The abnormal choline metabolism in cancer cells is closely related to tumor progression [20]. Abnormal choline uptake and choline phospholipid metabolism in GBM cells were confirmed by MRI and PET [30,31]. However, in this study, GPC administration had a beneficial association with the survival of GBM patients. 

In the current study, the GPC group (≥3 mos) demonstrated a slight increase in OS and PFS, compared to the non-GPC group (<3 mos). In short-term use, the effect of GPC on survival was limited, and the difference between the study and control groups was not prominent. Patients with long-term use of GPC (≥12 mos), however, showed a significant increase in OS, compared to the control group (<12 mos). The effect of long-term GPC use has also been shown in a study, evaluating changes in cognitive function following GPC use [21]. As a repurposed drug, GPC might exhibit minor improvements that are not readily observable during rapid clinical progression in patients with GBM. Well-known prognostic factors, such as the extent of resection, standard treatment using Stupp’s regimen, the methylation status of MGMT promoter, or age, were a major influence on patient survival. There is no doubt that the administration of GPC cannot exceed the effect of those prognostic factors on the survival of GBM patients. Nevertheless, in this study, GPC use was positively associated with patient survival even after the correction of confounding factors. GPC could be considered as an additional agent to the existing conventional treatment of patients with GBM, while future prospective controlled studies are required for verification of this positive correlation.

It is not clear how GPC affects the survival of patients with GBM. GBM cells are influenced by the tumor microenvironment [32]. The neuron is an important component of the glioma microenvironment and regulates tumor growth in an activity-dependent manner [33]. Glioma cells form synapses with neurons, the neuroglial synapses [34,35]. Studies on the role of ACh, a neurotransmitter, and its receptors in GBM have been limited and conflicting [17,18]. ACh receptors are upregulated in active infiltration zones in GBM [16]. When stimulated by ACh, nicotinic ACh receptors containing the a7 or a9 subunits increase the proliferation of GBM cells [36]. On the contrary, the activation of the M2 muscarinic receptor inhibits cell proliferation in GBM [18]. ACh receptor activation increases cell invasion but does not alter cell proliferation [16]. Non-selective activation of various ACh receptor subtypes can trigger opposite signaling pathways, allowing GPC, a choline precursor, to have minimal impact on cell fate. 

Glutamatergic excitations in neuroglial synapses promote GBM cell proliferation and invasion, causing tumor progression after binding to glutamate receptors [37,38]. The increase in cholinergic transmission induced by GPC may reduce glutamate neurotoxicity via activation of ACh receptors [39,40]. The use of GPC is hypothesized to reduce glutamatergic excitation, inhibiting tumor progression. Neural circuit effects on neuroglial synapse have not been studied much as therapeutic options for GBM treatment.

The current study has several limitations. There was a certain selection bias because of its retrospective nature. Although the baseline clinical characteristics between the study and control groups were not significantly different, several confounding factors may have influenced survival outcomes. To overcome this limitation, we conducted a multivariate analysis, including well-known prognostic factors, such as age, KPS score, extent of resection, and MGMT promoter methylation status. The multivariate analysis also showed that the use of GPC has a significant effect on OS in long-term usage. Another issue is that there was no data on cognitive improvement with GPC in GBM patients. The objective measurement of cognitive function was not performed in this study and should be evaluated in the future. Additionally, there has been a lack of evidence on the optimal dosage and frequency of GPC for survival benefits in GBM patients. The reason for the discontinuation of GPC was not fully acquired from the medical record. Despite the statistical significance, it could be an exaggeration to draw clear conclusions and recommendations on the survival of GPC. Prospectively designed and multi-center clinical trials should be needed to prove the role of GPC on the survival benefit of GBM patients. Since the underlying mechanisms of GPC in GBM are not fully understood, the role of GPC on GBM cells and microenvironments should be validated by basic research.

## 5. Conclusions

To the best of our knowledge, the current study is the first to assess the survival benefit of GPC in patients with GBM. In this retrospective analysis, the use of GPC was somewhat associated with prolonged OS and PFS in GBM patients. GPC can be added to the conventional, standard treatment for patients with newly diagnosed GBM. The neural circuit effects on survival require more large-scale clinical studies to establish as a therapeutic option for GBM patients.

## Figures and Tables

**Figure 1 jcm-11-06052-f001:**
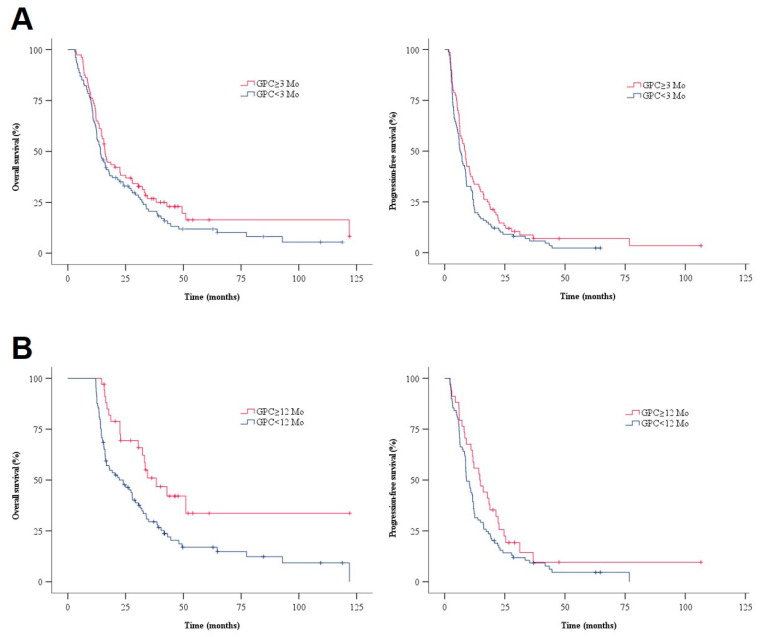
OS and PFS in the enrolled patients. (**A**) Kaplan–Meier survival curves showing the OS and PFS of patients according to the GPC use in all patients (*n* = 187). The median OS was 16.1 months in the group that used GPC (usage duration ≥3 mos, *n* = 80) and 14.2 months in the group that did not (usage duration <3 mos, *n* = 107) (*p* = 0.158). Median PFS was 8.4 months in the GPC group and 6.5 months in non-GPC group (*p* = 0.092). (**B**) Kaplan–Meier survival curves showing the OS and PFS of patients, according to the long-term use of GPC in patients followed for more than 12 months after initial diagnosis (*n* = 123). Median OS was 38.3 months in the long-term GPC group (usage duration ≥12 mos, *n* = 34) and 24.0 months in the non-long term GPC group (usage duration <12 mos, *n* = 89) (*p* = 0.004). The median PFS was 14.6 months in the long-term GPC group and 9.1 months in the non-long term GPC group (*p* = 0.082). (GPC, choline alphoscerate; OS, overall survival; PFS, progression-free survival).

**Table 1 jcm-11-06052-t001:** Patient demographics according to the use of GPC (≥3 mos) (*n* = 187).

Parameter	No. of Patients (%)			*p*-Value
	Total (*n* = 187)	GPC Group (*n* = 80)	Non-GPC Group (*n* = 107)	
Median age at diagnosis, yrs (IQR)	64.0 (54–72)	63.0 (54–70)	66.0 (53–74)	0.686
Gender (male)	101 (54)	42 (53)	59 (55)	0.768
Preoperative KPS score (mean ± SD)	80 ± 13	82 ± 11	79 ± 14	0.119
Postoperative KPS score (mean ± SD)	81 ± 10	83 ± 7	79 ± 12	0.033
Median tumor volume (cm^3^, IQR)	31.6 (16–57)	29.6 (15–49)	33.4 (16–62)	0.372
Location of tumor				0.014
Supratentorial	168 (90)	77 (96)	91 (85)	
Infratentorial	19 (10)	3 (4)	16 (15)	
Extent of resection				0.080
Gross total resection	79 (42)	39 (49)	40 (37)	
Subtotal resection	63 (34)	28 (35)	35 (33)	
Biopsy	45 (24)	13 (16)	32 (30)	
MGMT promoter methylation				0.878
Methylated	97 (56)	44 (57)	53 (56)	
Unmethylated	75 (44)	33 (43)	42 (44)	
Not available	15	3	12	
CCRT	130 (70)	68 (85)	62 (58)	<0.001
Median PFS, mos (95% CI)	7.2 (5.8–8.6)	8.4 (6.6–10.1)	6.5 (5.4–8.0)	0.092
Tumor relapse	177 (94)	74 (93)	103 (96)	
Median OS, mos (95% CI)	15.5 (14.1–17.0)	16.1 (14.6–17.6)	14.2 (11.8–16.6)	0.158
Death	154 (82)	62 (78)	92 (86)	

Abbreviations: CCRT, concomitant chemoradiotherapy; CI, confidence interval; GPC, choline alphoscerate; IQR, interquartile range; KPS, Karnofsky performance status; MGMT, O6-methylaguanine-DNA methyltransferase; mos, months; OS, overall survival; PFS, progression-free survival; SD, standard deviation; yrs, years.

**Table 2 jcm-11-06052-t002:** Prognostic factors for OS and PFS in the whole series (*n* = 187).

Factor	OS				PFS			
	Univariate		Multivariate		Univariate		Multivariate	
	*p*-Value	HR (95% CI)	*p*-Value	HR (95% CI)	*p*-Value	HR (95% CI)	*p*-Value	HR (95% CI)
Age	<0.001	1.046 (1.031–1.062)	<0.001	1.036 (1.018–1.054)	0.001	1.025 (1.010–1.039)	0.046	1.016 (1.000–1.033)
Preop. KPS	0.031	0.988 (0.978–0.999)	0.457	-	0.033	0.988 (0.978–0.999)	0.632	-
Postop. KPS	0.001	0.977 (0.964–0.990)	0.052	0.983 (0.967–1.000)	0.092	0.988 (0.975–1.002)	0.838	-
Supratentorial location	0.001	0.430 (0.260–0.711)	0.702		0.002	0.457 (0.281–0.744)	0.53	-
Complete resection	<0.001	0.466 (0.333–0.650)	<0.001	0.482 (0.337–0.725)	0.002	0.613 (0.452–0.830)	0.008	0.647 (0.469–0.892)
Methylated MGMT promoter	0.083	0.745 (0.535–1.039)	<0.001	0.507 (0.354–0.725)	0.012	0.669 (0.489–0.915	0.004	0.616 (0.443–0.855)
CCRT	<0.001	0.257 (0.180–0.366)	0.001	0.453 (0.286–0.719)	<0.001	0.365 (0.260–0.512)	0.002	0.526 (0.348–0.797)
GPC usage (≥3 mos)	0.159	0.792 (0.573–1.096)	0.799	-	0.094	0.774 (0.573–1.045)	0.886	-

Abbreviations: CCRT, concomitant chemoradiotherapy; CI, confidence interval; GPC, choline alphoscerate; KPS, Karnofsky performance status; MGMT, O6-methylaguanine-DNA methyltransferase; mos, months; PFS, progression-free survival; Preop, preoperative; Postop, postoperative; OS, overall survival.

**Table 3 jcm-11-06052-t003:** Patient demographics according to the long-term use of GPC (≥12 mos) (*n* = 123).

Parameter	No. of Patients (%)			*p*-Value
	Total (*n* = 123)	Long-Term GPC Group (*n* = 34)	Non-Long-Term GPC Group (*n* = 89)	
Median age at diagnosis, yrs (IQR)	59.0 (52–68)	59.0 (52–65)	59.0 (52–70)	0.561
Gender (male)	64 (52)	18 (53)	46 (52)	1.000
Preoperative KPS score (mean ± SD)	82 ± 12	83 ± 13	81 ± 12	0.192
Postoperative KPS score (mean ± SD)	83 ± 8	85 ± 6	82 ± 8	0.097
Median tumor volume (cm^3^, IQR)	30.0(15–57)	31.5 (16–52)	28.0 (15–58)	0.653
Location of tumor				0.105
Supratentorial	115 (93)	34 (100)	81 (91)	
Infratentorial	8 (7)	0	8 (9)	
Extent of resection				0.138
Gross total resection	63 (51)	20 (59)	43 (49)	
Subtotal resection	39 (32)	12 (35)	27 (30)	
Biopsy	21 (17)	2 (6)	19 (21)	
MGMT promoter methylation				1.000
Methylated	70 (60)	20 (61)	50 (60)	
Unmethylated	46 (40)	13 (39)	33 (40)	
Not available	7	1	1	
CCRT	105 (85)	33 (97)	72 (81)	0.042
Median OS, mos (95% CI)	27.8 (20.9–34.7)	38.3 (27.5–49.1)	24.0 (15.7–32.3)	0.004
Tumor relapse	90 (73)	18 (53)	72 (81)	
Median PFS, mos (95% CI)	11.0 (8.7–13.3)	14.6 (9.2–20.0)	9.1 (7.2–11.0)	0.082
Death	113 (92)	29 (85)	84 (94)	

Abbreviations: CCRT, concomitant chemoradiotherapy; CI, confidence interval; GPC, choline alphoscerate; IQR, interquartile range; KPS, Karnofsky performance status; MGMT, O6-methylaguanine-DNA methyltrans-ferase; mos, months; OS, overall survival; PFS, progression-free survival; SD, standard deviation; yrs, years.

**Table 4 jcm-11-06052-t004:** Prognostic factors for OS and PFS in patients followed for more than 12 months (*n* = 123).

Factor	OS				PFS			
	Univariate		Multivariate		Univariate		Multivariate	
	*p*-Value	HR (95% CI)	*p*-Value	HR (95% CI)	*p*-Value	HR (95% CI)	*p*-Value	HR (95% CI)
Age	0.002	1.031 (1.011–1.051)	<0.001	1.041 (1.020–1.064)	0.575	1.005 (0.988–1.022)	0.48	-
Preop. KPS	0.391	0.993 (0.978–1.009)	0.667	-	0.413	0.994 (0.979–1.009)	0.626	-
Postop. KPS	0.814	0.997 (0.975–1.020)	0.556	-	0.505	1.008 (0.985–1.031)	0.181	-
Supratentoral location	0.024	0.403 (0.183–0.885)	0.014	0.304 (0.118–0.787)	0.033	0.451 (0.216–0.939)	0.639	-
Complete resection	0.002	0.520 (0.342–0.792)	0.001	0.454 (0.289–0.715)	0.094	0.728 (0.502–1.056)	0.064	0.696 (0.475–1.021)
Methylated MGMT promoter	0.311	0.800 (0.520–1.231)	0.01	0.525 (0.321–0.859)	0.05	0.677 (0.458–1.000	0.065	0.692 (0.468–1.023)
CCRT	0.001	0.360 (0.207–0.627)	0.169	-	0.001	0.406 (0.241–0.684)	0.003	0.445 (0.263–0.752)
Long-term GPC usage UUU (≥ 12 mos)	0.005	0.477 (0.284–0.801)	0.019	0.532 (0.314–0.900)	0.085	0.689 (0.451–1.053)	0.358	-

Abbreviations: CCRT, concomitant chemoradiotherapy; CI, confidence interval; GPC, choline alphoscerate; KPS, Karnofsky performance status; MGMT, O6-methylaguanine-DNA methyltransferase; mos, months; PFS, progression-free survival; Preop, preoperative; Postop, postoperative; OS, overall survival.

## Data Availability

The datasets used and analyzed during the current study are available from the corresponding author on reasonable request.

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
