# Peer review of "Effect of Choline Alphoscerate on the Survival of Glioblastoma Patients: A Retrospective, Single-Center Study"

_jcm, 2022, doi:10.3390/jcm11206052_

Round 1

Reviewer 1 Report

This is a generally well-written study of the association between the use of Choline Alphoscerate (GPC) and OS and PFS in patients over 17 years of age from a single centre with IDH-wildtype GBM. The study is retrospective in nature and finds an association between long-term use (> 12 months) of GPC and prolonged OS. this finding is potentially confounded by other factors in the study and therefore will need corroboration in a prospectively designed trial but this paper addresses an interesting hypothesis. It would be useful to better describe the use of GPC in GBM treatment with better references and refer to the extent of its use in different settings across the world to make this study more generally applicable. The manuscript makes claims that cannot be made at times and the authors need to revise their discussion in the light of the methodological limitations of this type of retrospective study.

My major concerns are:

The statement on line 48 that "neuro-oncologists have prescribed GPC to those with GBM to improve cognitive functioning" needs additional citations, a brief review of the literature demonstrates very little evidence that this is routinely practised in a global setting. Please supply further evidence. I cannot access reference 11 but the abstract provides no evidence of this treatment approach for GBM.

The aim of the study could be better defined at the end of the introduction - a clearly defined research question or hypothesis should replace the comment: "Here, we assess the survival effect of GPC for patients with GBM" on line 58. There are obvious methodological limitations related to retrospective studies that mean this survival effect cannot be convincingly assessed based on this study design.

Line 201: Claims that this study aimed to "verify the effect of GPC on survival outcome in patients with GBM". This study was a retrospective analysis of patient data and therefore could not be designed to verify this effect - a prospective study would be needed. The language needs to change to reflect this e.g. softer language such as "this study was designed to investigate the effect of GPC on survival outcome..." would be more appropriate. 

Lines 220-21: "in this study, 220 GPC administration had a beneficial outcome on the survival of GBM patients." This study does not demonstrate a beneficial outcome as a result of GPC. This retrospective analysis shows an association between GPC and outcome, not a causative effect as the authors imply in the discussion. A similar claim is made on lines 233-235 and is claimed even more strongly to the extent that adding GPC to GBM treatment protocols is recommended - this study does not provide sufficient data to make this suggestion and the text needs to be revised to reflect this. 

Why were the short-term GPC group only treated for the short term? For example - were they prevented from taking the medication due to progressive disease or disability? Could they not afford the medication and therefore survival was confounded by lower socioeconomic status? This needs to be explored in the results and discussion sections of the text.

My minor comments are:

Line 88: Stupp's regimen should be accompanied by a citation.

Line 29: Keywords: Glioblastoma has a spelling error.

Author Response

October 10, 2022

Dear Editor-in-chief,

I’d like to thank the editor and reviewers of ‘Journal of Clinical Medicine’ for taking time to review our manuscript (1923698). We have made some corrections and clarifications in the manuscript after going over your comments. The changes are summarized below. (In the revised manuscript, edited sentences are highlighted by the 'track changes' function of MS word):

The 1st Reviewer’s points

This is a generally well-written study of the association between the use of Choline Alphoscerate (GPC) and OS and PFS in patients over 17 years of age from a single centre with IDH-wildtype GBM. The study is retrospective in nature and finds an association between long-term use (> 12 months) of GPC and prolonged OS.

  1. This finding is potentially confounded by other factors in the study and therefore will need corroboration in a prospectively designed trial but this paper addresses an interesting hypothesis. It would be useful to better describe the use of GPC in GBM treatment with better references and refer to the extent of its use in different settings across the world to make this study more generally applicable. The manuscript makes claims that cannot be made at times and the authors need to revise their discussion in the light of the methodological limitations of this type of retrospective study.

Response) Thank you for your constructive comment. The limitation of this study was described in the discussion section. As you addressed, we added and changed several sentences to strengthen this limitation in the discussion section.  

- The current study has several limitations. There was a certain selection bias because of its retrospective nature. Although the baseline clinical characteristics between the study and control groups were not significantly different, several confounding factors may have influenced survival outcomes. To overcome this limitation, we conducted a multivariate analysis including well-known prognostic factors such as age, KPS score, extent of resection, and MGMT promoter methylation status. The multivariate analysis also showed that the use of GPC has a significant effect on OS in long-term usage. Another issue is that there was no data on cognitive improvement with GPC in GBM patients. The objective measurement of cognitive function was not performed in this study and should be evaluated in the future. Also, there has been a lack of evidence on the optimal dosage and frequency of GPC for survival benefits in GBM patients. “The reason for the discontinuation of GPC was not fully acquired from the medical record. Despite the statistical significance, it could be an exaggeration to draw clear conclusions and recommendations on the survival of GPC. Prospectively designed & multi-center clinical trials should be needed to prove the role of GPC on the survival benefit of GBM patients. Since the underlying mechanisms of GPC in GBM are not fully understood, furthermore, the role of GPC on GBM cells & microenvironments should be validated by basic research.”

My major concerns are:

  1. The statement on line 48 that "neuro-oncologists have prescribed GPC to those with GBM to improve cognitive functioning" needs additional citations, a brief review of the literature demonstrates very little evidence that this is routinely practised in a global setting. Please supply further evidence. I cannot access reference 11 but the abstract provides no evidence of this treatment approach for GBM.

Response) Thank you for your constructive comment. In recent investigations for various clinical settings (Efficacy and Safety of the Association of Nimodipine and Choline Alphoscerate in the Treatment of Cognitive Impairment in Patients with Cerebral Small Vessel Disease. The CONIVaD Trial, Drugs Aging. 2021 Jun;38(6):481-491)(Quantitative electroencephalography changes in patients with mild cognitive impairment after choline alphoscerate administration. J Clin Neurosci. 2022 Aug;102:42-48.). GPC has been proved to have positive effect on cognitive improvement.    

As you pointed out, although the use of GPC during GBM treatment has not been established, many neuro-oncologists have worried about the declined of cognitive function caused by radiation treatment or surgical resection and prescribed several drugs (End-of-Life Care in High-Grade Glioma Patients. The Palliative and Supportive Perspective. Brain Sci. 2018 Jun 30;8(7):125.). In our country, GPC has often prescribed before and after brain surgery, as well as for degenerative brain diseases and more than 30,000 people/year were new-user of GPC. (Association of L-α Glycerylphosphorylcholine With Subsequent Stroke Risk After 10 Years, JAMA Netw Open. 2021) Even in the report on the prescription of GPC in our country, there is no available report about the brain tumor-specific prescription. We changed the sentences and added references to clear this points.

- “Since GPC is well tolerated with adequate central nervous system penetration [11], recent investigations have proved that GPC has some positive effects on cognitive improvement in various clinical settings [12-13]. Although the guideline for the pharmacological approach has not been established, many neuro-oncologists have worried about the decline of cognitive function caused by treatments or disease progression and prescribed several drugs [14]. In our country, a recent paper showed that GPC has often been prescribed for various medical situations, including after brain surgery [15]. Many neuro-oncologists in our country have also considered GPC for those with GBM to improve cognitive functioning.”

  1. The aim of the study could be better defined at the end of the introduction - a clearly defined research question or hypothesis should replace the comment: "Here, we assess the survival effect of GPC for patients with GBM" on line 58. There are obvious methodological limitations related to retrospective studies that mean this survival effect cannot be convincingly assessed based on this study design.

Response) Thank you for your constructive comments. As you recommended, we changed the sentence in the introduction section.

- “Starting with this question, we initiated a retrospective study to evaluate whether there is a survival benefit of GPC in patients with GBM.”

  1. Line 201: Claims that this study aimed to "verify the effect of GPC on survival outcome in patients with GBM". This study was a retrospective analysis of patient data and therefore could not be designed to verify this effect - a prospective study would be needed. The language needs to change to reflect this e.g. softer language such as "this study was designed to investigate the effect of GPC on survival outcome..." would be more appropriate.

Response) Thank you for your constructive comments. As you recommended, we changed the sentence in the discussion section.

- “This study was designed to investigate the effect of GPC on survival outcome in patients with GBM.”

  1. Lines 220-21: "in this study, 220 GPC administration had a beneficial outcome on the survival of GBM patients." This study does not demonstrate a beneficial outcome as a result of GPC. This retrospective analysis shows an association between GPC and outcome, not a causative effect as the authors imply in the discussion. A similar claim is made on lines 233-235 and is claimed even more strongly to the extent that adding GPC to GBM treatment protocols is recommended - this study does not provide sufficient data to make this suggestion and the text needs to be revised to reflect this.

Response) Thank you for your constructive comments. As you recommended, we changed the sentences in the discussion section.

- Line 220-221: “However, in this study, GPC administration had a beneficial association with the survival of GBM patients.”

- Line 233-235: “Nevertheless, in this study, GPC use was positively associated with patient survival even after the correction of confounding factors. GPC could be considered as an additional agent to the existing conventional treatment of patients with GBM, while future prospective controlled studies are required for verification of this positive correlation.”

  1. Why were the short-term GPC group only treated for the short term? For example - were they prevented from taking the medication due to progressive disease or disability? Could they not afford the medication and therefore survival was confounded by lower socioeconomic status? This needs to be explored in the results and discussion sections of the text.

Response) Thank you for your constructive comments. Many factors such as disease progression, lower socioeconomic status, even whim can be a reason for drug cessation. As a limitation of the retrospective study, there were no specific medical records for the reason for cessation. We added the sentence in the discussion section (limitation of the study).

- Another issue is that there was no data on cognitive improvement with GPC in GBM patients. The objective measurement of cognitive function was not performed in this study and should be evaluated in the future. Also, there has been a lack of evidence on the optimal dosage and frequency of GPC for survival benefits in GBM patients. “The reason for the discontinuation of GPC was not fully acquired from the medical record.”

My minor comments are:

  1. Line 88: Stupp's regimen should be accompanied by a citation.

Response) Thank you for your constructive comments. We added a reference (Radiotherapy plus concomitant and adjuvant temozolomide for glioblastoma. N Engl J Med. 2005 Mar 10;352(10):987-96.).

  1. Line 29: Keywords: Glioblastoma has a spelling error.

Response) Thank you for your constructive comments. We corrected a typo.

Hopefully the revised manuscript will better meet the requirements of the ‘Journal of Clinical Medicine’ for publication. I’d like to thank you again for the constructive comments by reviewers.

Sincerely yours,

Kyung-Sub Moon, M.D., Ph.D.

Professor,

Department of Neurosurgery, Chonnam National University Hwasun Hospital and Medical School, 322 Seoyang-ro, Hwasun-eup, Hwasun-gun, Jeollanam-do, 519-763, South Korea

Tel:+82-61-379-7666, Fax:+82-61-379-7673, e-mail: moonks@chonnam.ac.kr

Reviewer 2 Report

The study is interesting and sound, although the finding is negative. Younger age, supratentorial location, complete resection, and MGMT promoter methyl (TMZ use) is already known to significantly associate with longer PSF and OS. The authors need to isolate the effect of GPC use from these variables, to find out if there is a true effect on survival by >12 month GPC use.

In the study cohort, the GPC group is somewhat younger. More importantly, the percent of patients with gross total resection and with supratentorial location in significantly higher in the GPC group compared to the non-GPC group. Also, the GPC group had to a significantly larger extent gone through complete treatment with concomitant chemoradiotherapy, which is known to effect OS.

The analysis to show benefits of >12-month GPC use is not convincing, and therefore overstated in the abstract.

Author Response

October 10, 2022

Dear Editor-in-chief,

I’d like to thank the editor and reviewers of ‘Journal of Clinical Medicine’ for taking time to review our manuscript (1923698). We have made some corrections and clarifications in the manuscript after going over your comments. The changes are summarized below. (In the revised manuscript, edited sentences are highlighted by the 'track changes' function of MS word):

2nd reviewer’s points

  1. The study is interesting and sound, although the finding is negative. Younger age, supratentorial location, complete resection, and MGMT promoter methyl (TMZ use) is already known to significantly associate with longer PSF and OS. The authors need to isolate the effect of GPC use from these variables, to find out if there is a true effect on survival by >12 month GPC use.

Response) Thank you for your constructive comments. As you pointed out, there were several confounding factors to lead major limitation of this study. To overcome this limitation, we performed multivariate analysis using the Cox-proportional hazard model. We added and changed several sentences to strengthen this limitation in the discussion section.

- The current study has several limitations. There was a certain selection bias because of its retrospective nature. Although the baseline clinical characteristics between the study and control groups were not significantly different, several confounding factors may have influenced survival outcomes. To overcome this limitation, we conducted a multivariate analysis including well-known prognostic factors such as age, KPS score, extent of resection, and MGMT promoter methylation status. The multivariate analysis also showed that the use of GPC has a significant effect on OS in long-term usage. Another issue is that there was no data on cognitive improvement with GPC in GBM patients. The objective measurement of cognitive function was not performed in this study and should be evaluated in the future. Also, there has been a lack of evidence on the optimal dosage and frequency of GPC for survival benefits in GBM patients. “The reason for the discontinuation of GPC was not fully acquired from the medical record. Despite the statistical significance, it could be an exaggeration to draw clear conclusions and recommendations on the survival of GPC. Prospectively designed & multi-center clinical trials should be needed to prove the role of GPC on the survival benefit of GBM patients. Since the underlying mechanisms of GPC in GBM are not fully understood, furthermore, the role of GPC on GBM cells & microenvironments should be validated by basic research.”

  1. In the study cohort, the GPC group is somewhat younger. More importantly, the percent of patients with gross total resection and with supratentorial location in significantly higher in the GPC group compared to the non-GPC group. Also, the GPC group had to a significantly larger extent gone through complete treatment with concomitant chemoradiotherapy, which is known to effect OS.

Response) Thank you for your constructive comments. As you pointed out, major influencing factors, such as age, location, the extent of resection, and concomitant chemoradiotherapy showed differences in survival between groups. Although we performed multivariate analysis using the Cox-proportional hazard model, the limitation from the retrospective nature could not be overcome. As mentioned in discussion section, future prospectively designed & multi-center clinical trials should be needed to overcome the limitation and prove the role of GPC on the survival benefit of GBM patients. We added and changed several sentences to strengthen this limitation in the discussion section, as previously mentioned above.

  1. The analysis to show benefits of >12-month GPC use is not convincing, and therefore overstated in the abstract.

Response) Thank you for your constructive comments. This retrospective analysis shows an association between GPC and outcome, not a causative effect. We added and changed several sentences to strengthen this limitation in the discussion section, as previously mentioned above. And as you recommended, we changed the sentence in the abstract.

- “Despite the limitations of this study, long-term GPC use was possibly associated with prolonged survival in GBM patients. Multi-center prospective randomized studies with a large number of patients are needed to validate these findings.”

Hopefully the revised manuscript will better meet the requirements of the ‘Journal of Clinical Medicine’ for publication. I’d like to thank you again for the constructive comments by reviewers.

Sincerely yours,

Kyung-Sub Moon, M.D., Ph.D.

Professor,

Department of Neurosurgery, Chonnam National University Hwasun Hospital and Medical School, 322 Seoyang-ro, Hwasun-eup, Hwasun-gun, Jeollanam-do, 519-763, South Korea

Tel:+82-61-379-7666, Fax:+82-61-379-7673, e-mail: moonks@chonnam.ac.kr

Round 2

Reviewer 1 Report

The authors have adressed my comments in full and I have no additional comments.